# Insight into Sources of Benzene, TCE, and PFOA/PFOS in Groundwater at Naval Air Station Whiting Field, Florida, through Numerical Particle-Tracking Simulations

**Eric D. Swain** [1,*], **James E. Landmeyer** [2], **Michael A. Singletary** [3] **and Shannon E. Provenzano** [3]

1   U.S. Geological Survey, Caribbean-Florida Water Science Center, Davie, FL 33314, USA
2   U.S. Geological Survey, South Atlantic Water Science Center, Columbia, SC 29210, USA; jlandmey@usgs.gov
3   Naval Facilities Engineering Command, Jacksonville, FL 32212, USA; michael.a.singletary@navy.mil (M.A.S.); shannon.e.provenzano.civ@us.navy.mil (S.E.P.)
\*   Correspondence: edswain@usgs.gov

**Abstract:** Past waste-disposal activities at Naval Air Station Whiting Field (NASWF) have led to elevated concentrations of contaminants in the underlying sand and gravel aquifer. Contaminants include two of the most commonly detected chemicals in groundwater in many countries (benzene and trichloroethylene (TCE)) and the "forever chemicals" per- and poly-fluoroalkyl substances (PFAS) such as perfluorooctanoic acid (PFOA) and perfluorooctane sulfonic acid (PFOS). A MODFLOW model (the Whiting Field Groundwater Model (WFGM)) was previously developed for NASWF and the surrounding area to simulate groundwater flow. To obtain insight into groundwater flow pathways for the identification of potential source areas, the MODPATH particle-tracking application was applied to the WFGM for three public supply wells and three monitoring wells at NASWF. The travel time to recharge areas was estimated using concentrations of the groundwater age-dating solutes tritium (as helium ingrowth) and chlorofluorocarbons detected in the monitoring wells. Simulated travel times agree with the groundwater ages and indicate that the calibrated WFGM reasonably represents groundwater flow velocities and pathways. The MODPATH simulations confirm suspected on-base source areas to explain chemical detection in the monitoring wells. In contrast, the particle-tracking simulations indicate that potential source areas to the public supply wells include both on- and off-base sources. This is important because PFAS chemicals can have multiple sources, including land application of sludge-based fertilizers. This approach that combines groundwater age dating with particle-tracking simulations can be applied at similar sites characterized by benzene-, TCE-, and PFAS-contaminated groundwater.

**Keywords:** groundwater; contaminant transport; particle tracking; wells

## 1. Introduction

Naval Air Station Whiting Field (NASWF) is a U.S. Navy Base near Milton, Florida (Figure 1), that has been used for airplane and helicopter flight instruction since 1943 [1]. A variety of chemicals were used in support of these operations, including paint-stripping compounds, cleaning solvents, alkaline cleaners, detergents, mineral spirits, methyl ethyl ketone, isopropyl alcohol, oils, hydraulic fluids, and aqueous fire-fighting foams (AFFFs) that contain per- and poly-fluoroalkyl substances (PFASs) such as perfluorooctanoic acid (PFOA) and perfluorooctane sulfonic acid (PFOS). Waste disposal for some chemicals was onsite in disposal pits, dry wells, landfills, and waste-oil bowsers [2].

Between 2015 and 2020, a groundwater model was developed for the NASWF study area for the purpose of creating a digital conceptual site model (CSM) to learn more about aquifer conditions, groundwater flow, and contaminant transport [3]. A CSM is required by the U.S. Environmental Protection Agency (EPA) Superfund process to synthesize disparate data collected during site studies. The model, called the Whiting Field Groundwater Model (WFGM)

(WFGM), was developed and reported in [3] and has been shown to reasonably reproduce groundwater levels in the sand and gravel aquifer. It has been used to gain insight into the potential directions of groundwater flow. As such, the model is deemed useful for estimating the transport of contaminants from known (and unknown) source areas. To confirm or refute these known source areas and estimate potential source areas for PFAS chemicals, a method was used to couple the groundwater flow model with groundwater particle-tracking simulations, where the flow velocities simulated by the numerical model were used to move hypothetical particles through the sand and gravel aquifer. When performed forwards in time, particle tracking predicts the destination of solutes (such as contaminants) and, when performed backwards in time, particle tracking can suggest the potential origin(s) of solutes. The extent of travel was thus compared to groundwater age dates for three monitoring wells screened in the sand and gravel aquifer. This method was used to confirm or refute potential sources of contaminants.

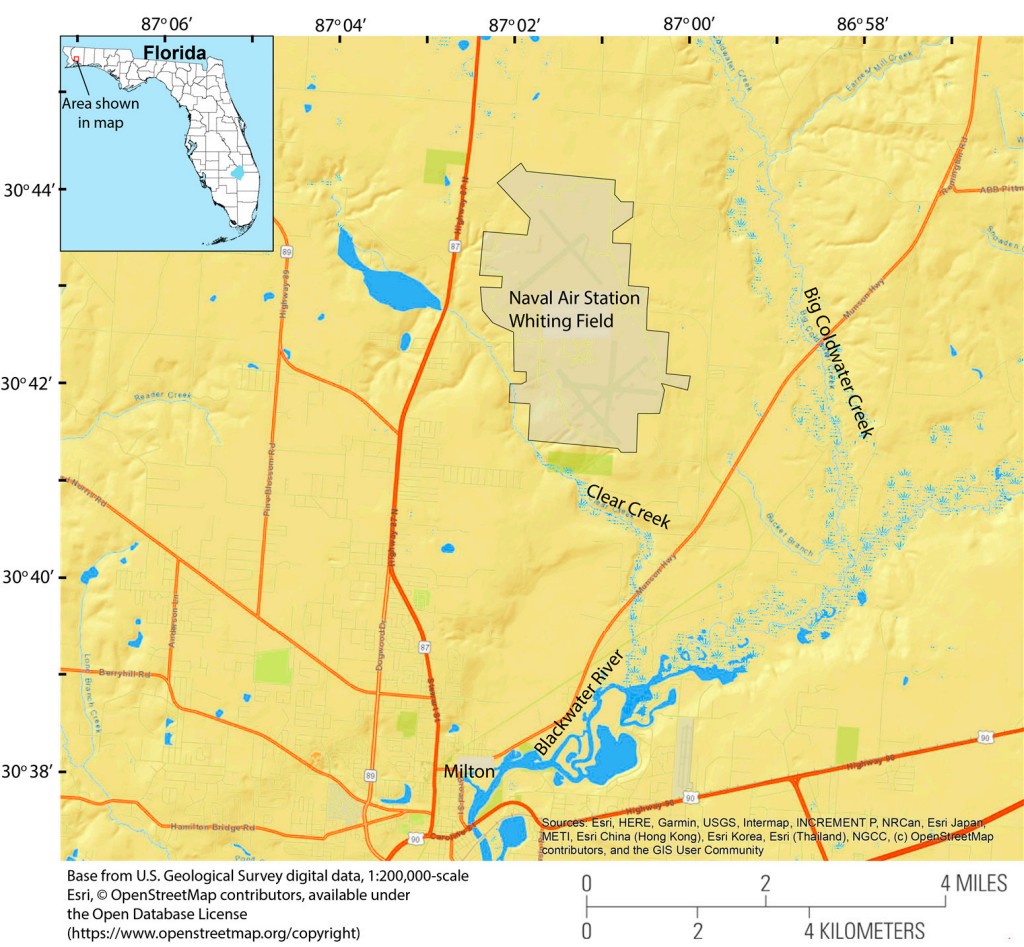

**Figure 1.** Location of Naval Air Station Whiting Field, near Milton, Florida.

Particle-tracking simulation for the WFGM has a special advantage in isotope age-dating data collected at wells in the study area. The comparison of these allowed for an evaluation of the travel times and supports the modeled travel distances. The particle-tracking results led to a reevaluation and expansion of the area considered as potentially contributing to contamination at test well locations. Previous to the particle-tracking simulation, only sources within Whiting Field were considered. This indicates that the particle-tracking technique applied can delineate contribution areas more quantitatively and help determine approaches to contaminant management.

The isotope age-dating techniques in NASWF are described in [3]. Similar techniques have been widely used in groundwater systems, not only as simple tracers but also to

identify constituents in groundwater and surface-water/groundwater mixing characteristics [4,5]. Isotope age dating in NASWF does not identify sources but can be used for travel time comparisons with particle tracking.

Currently, there are particle-tracking algorithms developed for unstructured model grids, limited to smoothed, rectangular-based quadtree and quadpatch grids [6]. Another particle-tracking algorithm has been developed for MODFLOW for unstructured grids and has been demonstrated for cases where interpolation between model cells is difficult due to high spatial variability in groundwater flow [7]. A particle-tracking algorithm has been developed for a finite-element groundwater model in two dimensions that considers the interpolated velocities within the model elements [8]. This was compared with MODPATH, which uses the finite-difference MODFLOW groundwater model.

Previous applications of particle tracking to groundwater models of field sites include an application to a wellfield in an urban area of Bordeaux, France, to estimate aquifer parameters for a contaminant transport problem with heads and concentrations combined in a weighted hybrid objective function [9]. Calibrating the groundwater model with particle-tracking results is somewhat of an inverse approach compared to what is done in our article here. A model of the western Lake Taupo catchment in New Zealand was used to compare particle tracking with solute transport simulations [10]. Comparing tritium concentrations computed by the solute transport model with particle tracks indicated similar travel times, but the particle distribution differed substantially from the solute distribution. In the Nägelstedt catchment in central Germany [11], particle tracking was developed for a groundwater model coupled to surface water. An analysis of sensitivity to model input parameters indicated that topography and aquifer characteristics can be important to particle-tracking results. Particle tracking in a groundwater model of southern Ghana was used by [12] to determine travel times from recharge sites in rural areas. The study did not have isotope age-dating data for comparison, as we have in this study.

This paper presents the use of a particle-tracking algorithm (MODPATH) in the groundwater model of NASWF to provide information on sources of contaminants detected in monitoring wells and public supply wells at NASWF. The groundwater model is summarized, the particle tracking is described along with the groundwater age-dating results, and the results are discussed. The particle tracking is discussed in the context of known contaminant detections at NASWF and the effect of public-supply well pumpage on contaminant transport in the sand and gravel aquifer.

## 2. Study Area

NASWF is located in Santa Rosa County, in the panhandle of Florida's northwestern coastal area, 8.9 km north of the city of Milton (Figure 1), and encompasses 15.55 square kilometers including two airfields (north field and south field) separated by various structures that support flight operations and staff (Figure 1). The area around NASWF includes fertilized agricultural land to the northwest, residential and forested areas with some fertilized agricultural land to the south and southwest, and forests to the east [3]. Ground elevation in the NASWF area ranges from 46 to 58 m above mean sea level (amsl). To the west, NASWF is partially bounded by Clear Creek with an average altitude of about 12 m amsl. Clear Creek, and the meandering Big Coldwater Creek to the east, are tributaries of the Blackwater River (Figure 1). The Florida Department of Environmental Protection classifies Clear Creek and Big Coldwater Creek as Class III waters for recreation, propagation, and the management of fish and wildlife. Blackwater River is classified as Outstanding Florida Water [13].

The sand and gravel aquifer at NASWF is composed of unconsolidated Holocene and Pleistocene alluvium and terrace deposits, the Citronelle Formation, and unnamed clastics of the upper Miocene age [14]. A generalized stratigraphic column is shown in Figure 2. Groundwater is present under perched to water table conditions. A groundwater divide exists at NASWF such that groundwater recharge on the western side flows to the southwest to discharge to Clear Creek and groundwater recharge on the eastern side flows to the southeast to discharge to Big Coldwater Creek.

| Series | Stratigraphic and hydrologic units | | | Lithology |
|---|---|---|---|---|
| Holocene and Pleistocene | Alluvium and terrace deposits | | Sand and gravel aquifer | Undifferentiated silt, sand, and gravel with some clay. Surficial zone of aquifer. |
| Pliocene | Citronelle Formation | | | Sand, very fine to very coarse and poorly sorted. Hardpan layers in upper part. |
| Miocene | Unnamed coarse clastics | Shoal River Formation | | Sand, shell, and marl. |
| | | Alum Bluff Group Shoal River Formation Chipola Formation | | Sand with lenses of silt, clay, and gravel (includes unnamed coarse clastics and Alum Bluff Group). Main producing zone of aquifer. |
| | Pensacola Clay | | Confining unit | Dark to light gray sandy clay. Is basal confining unit in southern one-half of area. |
| | St. Marks Formation | | Floridan aquifer system | Limestone and dolomite composing the top of the Floridan aquifer system |

**Figure 2.** Generalized stratigraphic column, Naval Air Station Whiting Field, near Milton, Florida (modified from [14]).

In the 1980s, benzene and trichloroethylene (TCE) were detected in samples of potable water pumped from two of the three on-base potable wells, and it was determined that past waste disposal at NASWF had led to contamination of the unsaturated zone and the development of plumes in the underlying sand and gravel aquifer [2]. Since then, the groundwater contaminants have been delineated into a north-central plume and a south-central plume [3]. The north-central plume is characterized by chlorinated solvents and fuels that come from disposal areas at the land surface. The south-central plume is characterized by similar contaminants that come from other disposal areas at the land surface. The groundwater contamination is routinely monitored by the U.S. Navy Installation Restoration Program [3]. More recently, the "forever chemicals" of PFOA and PFOS have been detected above reported levels in raw groundwater from the public supply wells. The occurrence of potential on-base or off-base sources of these contaminants is the focus of ongoing monitoring by the Navy. Groundwater pumped from the public supply wells has been treated by on-site granulated activated carbon filters since 1987.

### 3. Materials and Methods

The U.S. Geological Survey (USGS) has developed the WFGM, a groundwater model application for NASWF and the surrounding area [3]. The WFGM application uses the MODFLOW-NWT code, which uses a Newton formulation of MODFLOW-2005 [15] to improve the computation of unconfined groundwater flow in a three-dimensional grid while representing hydrologic factors such as precipitation, evapotranspiration, ground–water/surface–water interactions, and well pumpage. MODFLOW can be used with MODPATH, a postprocessing program that uses the cell-by-cell flow data from MODFLOW to construct the groundwater velocity distribution for particle-tracking calculations [16].

*3.1. Monitoring and Public Supply Wells*

Six wells within NASWF were included in the particle-tracking analysis. These included three public supply wells (W-S2, W-W3, and W-N4) at higher altitudes near the center of NASWF, which were installed between 1943 and 1952 (Figure 3). The three monitoring wells included well WHF-15-MW-4S, which was sampled by the USGS in 2015. In brief, before sample collection, groundwater was pumped through a low-flow chamber, and measurements of the physical properties and chemical constituents of the groundwater, such as dissolved oxygen, pH, specific conductance, and temperature, were measured using a YSI 6920 sonde (YSI, Inc.). The sonde was calibrated daily before sampling. Groundwater samples were collected after measurements of dissolved oxygen, pH, specific conductance, and temperature had stabilized. Groundwater did not require filtration because of the low sample turbidity [3]. The chlorofluorocarbon (CFC) compound trichlorofluoromethane (CFC-11) values indicated that the water from the wells had recharged from the surface in 1976 [3]; the CFC-11 results of WHF-15-MW-5D farther to the south indicated a similar recharge date of 1977 [3]. An additional monitoring well, WHF-16-MW-7D, is located farther downgradient and closer to Clear Creek, and, in 1998, benzene was detected in this well. A groundwater age of 1973 was estimated for this well using tritium ($^3$H) and its daughter product of helium ($^3$He) concentrations during a previous sampling event in 1998 [3]. This dating information provides an independent travel time estimate for comparison with the particle-tracking simulation. Model parameters, such as porosity, could be calibrated with these measured travel times, but this was not attempted in this study. Information on the six wells used in the particle-tracking experiments is given in Table 1. The groundwater sample collection that occurred for these monitoring wells in 2015 indicated that the groundwater was oxic (3.93 and 6.35 milligrams per liter (mg/L)), acidic (4.73 and 4.56) and of low specific conductance (34 and 24 microsiemens per centimeter (μs/cm)) for WHF-15-MW-4S and WHF-15-MW-5D, respectively [3].

The primary questions of interest from the particle tracking included the following: (1) What are the source locations (recharge) for groundwater at a particular well? (2) What are the times of travel from a potential source to a particular well? The travel times estimated using CFC-11 from recharge to wells WHF-15-MW-4S and WHF-15-MW-5D were 39 and 38 years, respectively [3]. The estimated travel time from source to well WHF-16-MW-7D was 25 years [3]. In summary, these independently derived groundwater age dates provided calibration targets to compare to particle-tracking results, the distance of each of which was dependent on the groundwater flow rate. As such, this provided greater control and reduced uncertainty in confirming or refuting known source areas. More importantly, this approach provided a way to assess the validity of the locations of unknown sources.

**Table 1.** Wells included in MODPATH particle tracking; bls, below land surface; lpm, liters per minute; WFGM, Whiting Field Groundwater Model grid location [3].

| Well Name (Figure 3) | USGS ID | Easting | Northing | Well Type | Screened Depth m bls | Pumping Rate lpm | WFGM Row | Column |
|---|---|---|---|---|---|---|---|---|
| WHF-16-MW-7D | 304153087015101 | 497052 | 3396147 | Monitoring | - | - | 322 | 201 |
| WHF-15-MW-4S | 304147087012301 | 499488 | 3395483 | Monitoring | 28.7–33.3 | - | 333 | 223 |
| WHF-15-MW-5D | 304141087013201 | 498689 | 3394910 | Monitoring | 36.0–39.0 | - | 339 | 216 |
| W-N4 | 304244087010701 | 498222 | 3397722 | Public supply | 61.0–67.1 | 1900 | 276 | 237 |
| W-W3 | 304235087010201 | 498351 | 3397447 | Public supply | 51.8–61.3 | 1900 | 285 | 241 |
| W-S2 | 304225087005601 | 498515 | 3397120 | Public supply | 51.8–68.6 | 1900 | 295 | 247 |

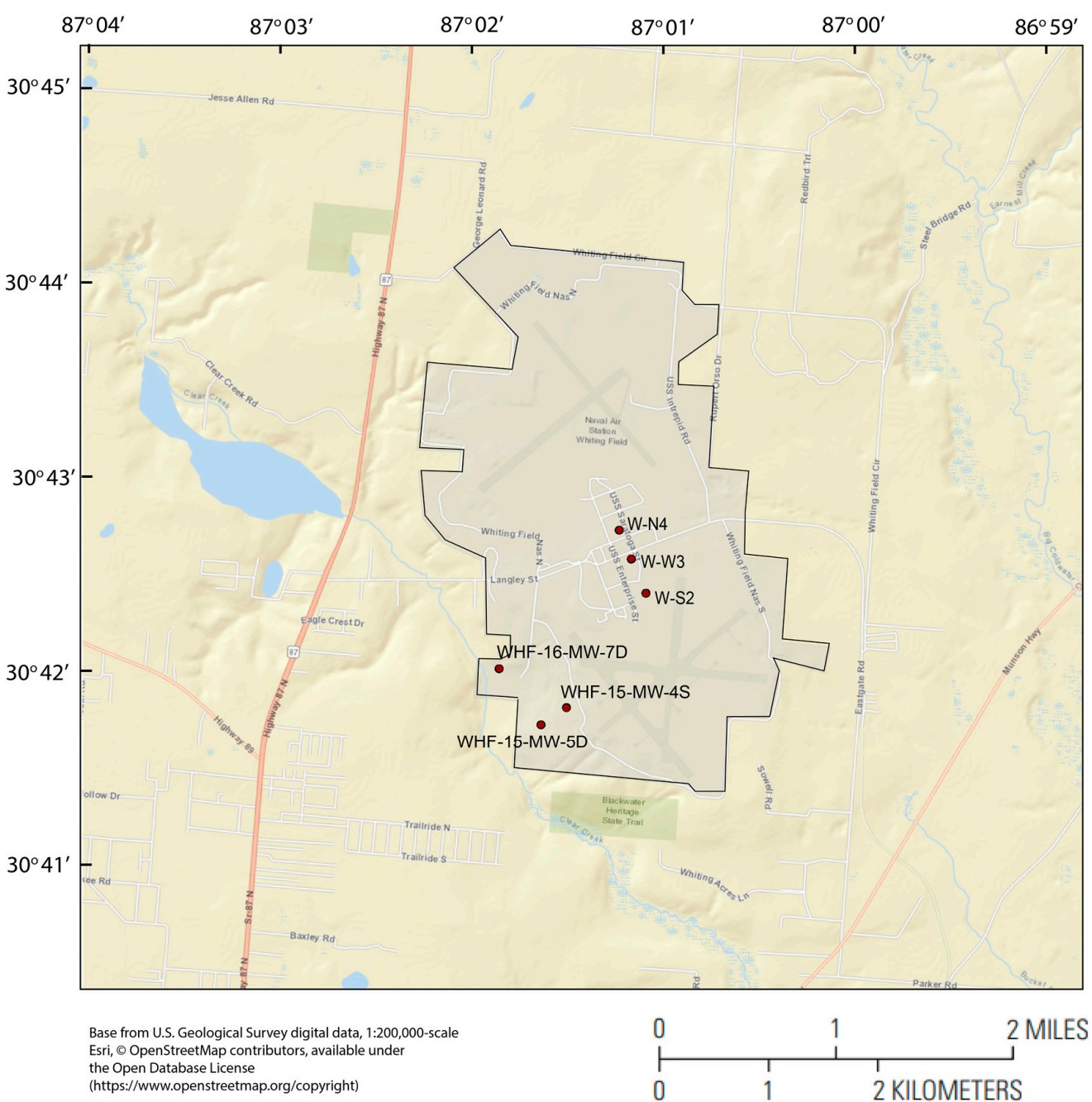

**Figure 3.** Locations of monitoring [WHF-n] and public supply [W-n] wells used in the particle-tracking simulations, where n is the specific well name.

### 3.2. Whiting Field Groundwater Model (WFGM)

Following is a general description of the WFGM; further details on the model development, calibration, and results can be found in [3]. The WFGM application of MODFLOW-NWT was developed for a 210-square-mile area approximately centered on NASWF (Figure 3). The WFGM's grid discretization of 30.5 ft in both horizontal directions yields 533 rows and 424 columns (Figure 4). Vertically, the WFGM contains 9 layers, with layers 2 through 8 each 15.2 m thick and layer 9 30.5 m thick. Due to the variations in land elevation, grid layers 1, 2, 3, and 4 of the groundwater model can have diminished thickness, and layers 1, 2, and 3 do not exist in parts of the WFGM domain. The lowest land altitude in the model area is one meter above the North American Vertical Datum of 1988.

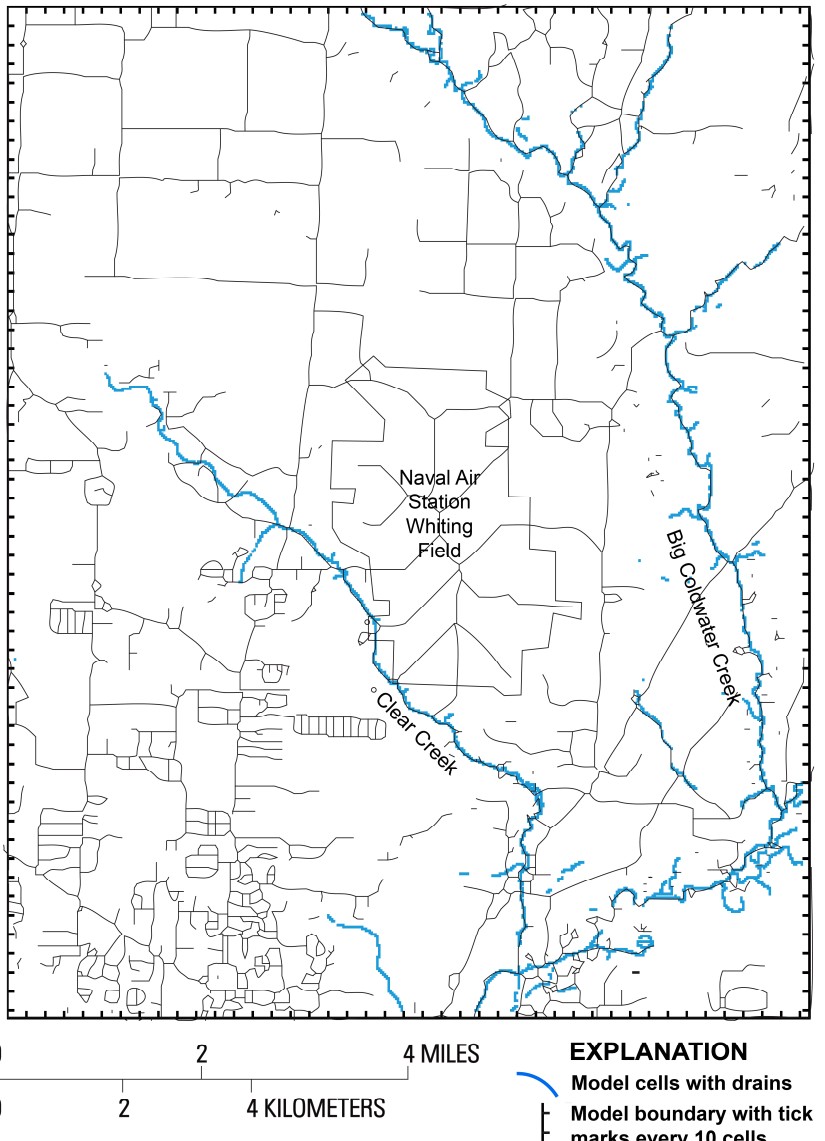

**Figure 4.** Model area with grid spacing and stream discretization.

The surface water system in the WFGM is dominated by Clear Creek and Big Coldwater Creek (Figure 3). These features are represented in MODFLOW using the Drain Package. The Drain Package was considered appropriate because the creeks in the area act as sinks for groundwater. Topographic coverage was used to generate creek locations. Control altitudes were defined by estimated average water levels in the creeks or, for dry times, the altitude of the creek bed, and the drain conductance was calibrated along with the aquifer hydraulic conductivity.

Recharge to the sand and gravel aquifer was simulated as precipitation minus evapotranspiration. Groundwater head measurements in the area indicated that the groundwater head was largely from 15.2 to 30.5 m bls, far more than nominal evapotranspiration extinction depths. Although evapotranspiration from the water table is considered negligible at the depths common in the WFGM area, some losses from interception storage and evapotranspiration in the unsaturated zone as the water percolates downward must be considered, as well as evapotranspiration from perched zones in upland areas and shallow water tables in riparian areas near creeks. The net recharge was calibrated for groundwater heads and flow measurements at a creek station [3], and the difference with known mean rainfall was considered to reflect interception storage and evapotranspiration.

With limited information available, porosities were set at a spatially uniform 0.3 value. In a steady-state simulation, porosity does not enter the groundwater head and flow computation. But, porosity is a significant part of particle-tracking computations due to its inverse relationship to pore velocities.

Calibration was advanced by using lithologic logs (Figure 5) to gain insight into the distribution of hydraulic conductivity and matching measured to simulated groundwater heads [3]. The lithologic type of each core was paired with the test model input hydraulic conductivity in the corresponding model cell to combine the information in the lithologic and test model input hydraulic conductivity arrays. The hydraulic conductivity magnitudes corresponding to each lithologic type were then used to make hydraulic conductivity adjustments. This method was a simple adjustment linearly interpolated between locations. More sophisticated techniques such as the generalized parameterization method [17] or hierarchical Bayesian model averaging [18] can produce more detailed hydrogeologic information and insight into aquifer structure, given sufficient data. However, the lithologic logs shown in Figure 5 were distributed so nonuniformly that a simple interpolation was considered appropriate.

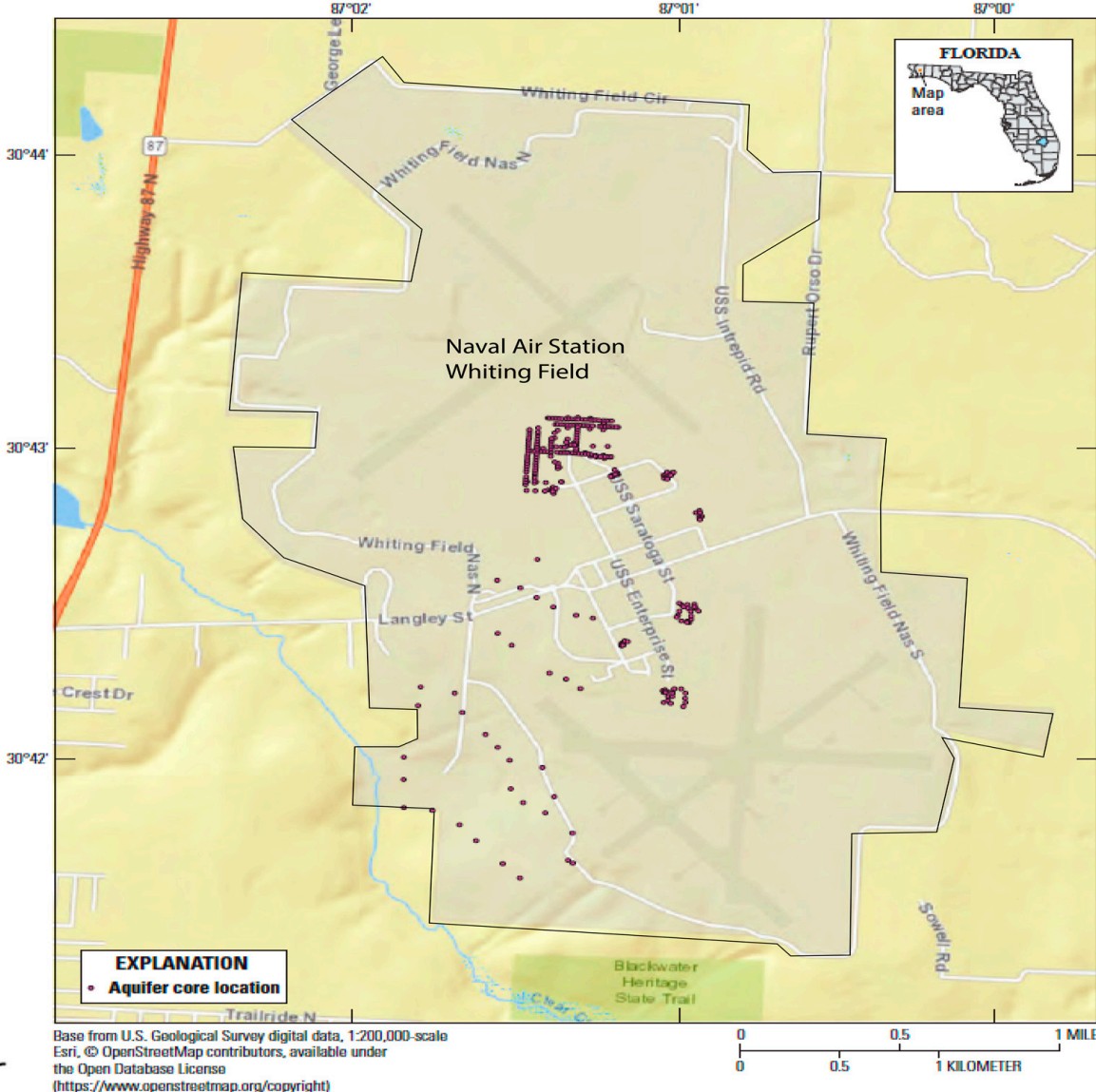

**Figure 5.** Locations of lithologic cores in the Whiting Field groundwater model area, Naval Air Station Whiting Field, near Milton, Florida (from [3]).

Groundwater head data for calibration were obtained from 59 previously existing monitoring wells in the study area (Figure 6). These wells were installed between 1993 and 1997 as part of the U.S. Navy's investigation of groundwater at NASWF. For the WFGM, 110 groundwater head measurements made at these monitoring wells during this period were used [3]. These wells are clustered in the central and southwestern parts of Naval Air Station Whiting Field, where the transport of contaminants towards Clear Creek is of most interest and the groundwater heads make ideal calibration targets. Discrete flow measurements made at three locations on Clear Creek (Figure 6) were compared to leakage rates simulated by the WFGM.

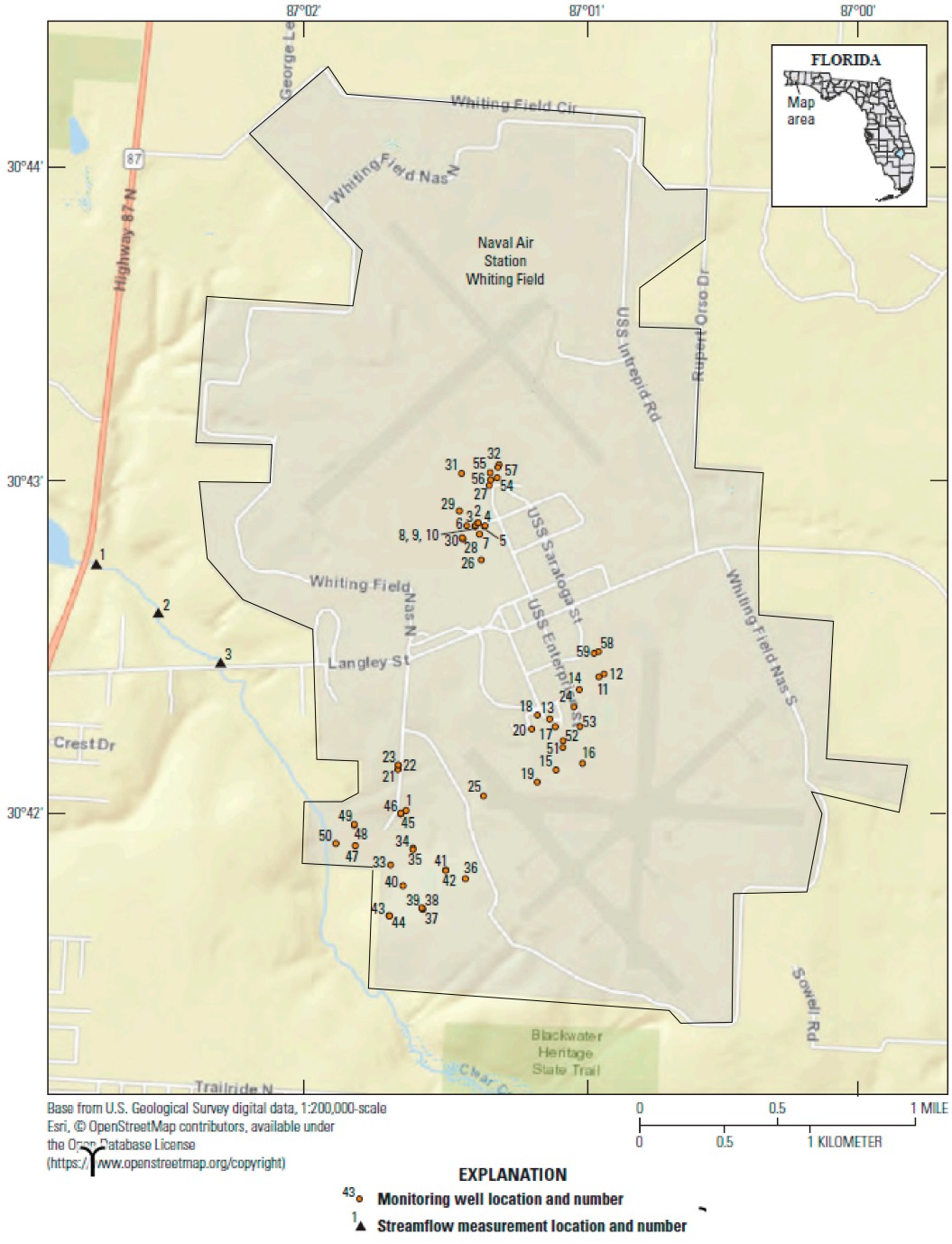

**Figure 6.** Location of monitoring wells used for calibration of the Whiting Field groundwater model. Also shown are the locations of the streamflow discharge measurements (from [3]). The numbers are identifiers explained in [3].

The WFGW simulated groundwater contours indicated that the aquifer beneath Naval Air Station Whiting Field ultimately flows to either Clear Creek or Big Coldwater Creek and to Blackwater River south of Naval Air Station Whiting Field (Figure 7). Clearly, the local surface water streams are a primary groundwater sink, as indicated by their distinct effect on the simulated groundwater potentiometric contours. Clear Creek is closest to Naval Air Station Whiting Field and causes the most marked effect on groundwater potentiometric contours in its southwest corner. Simulated flow vectors indicate the contaminant plume moving from source areas at higher altitudes towards lower altitudes to the southwest, but the application of particle tracking allowed for a more definitive estimate of contaminant transport sources and destinations, especially for those wells with age dates that could constrain flow pathway extents. All the necessary input and output files can be downloaded from [19]. The grid locations of the wells used in the particle tracking are given in Table 1.

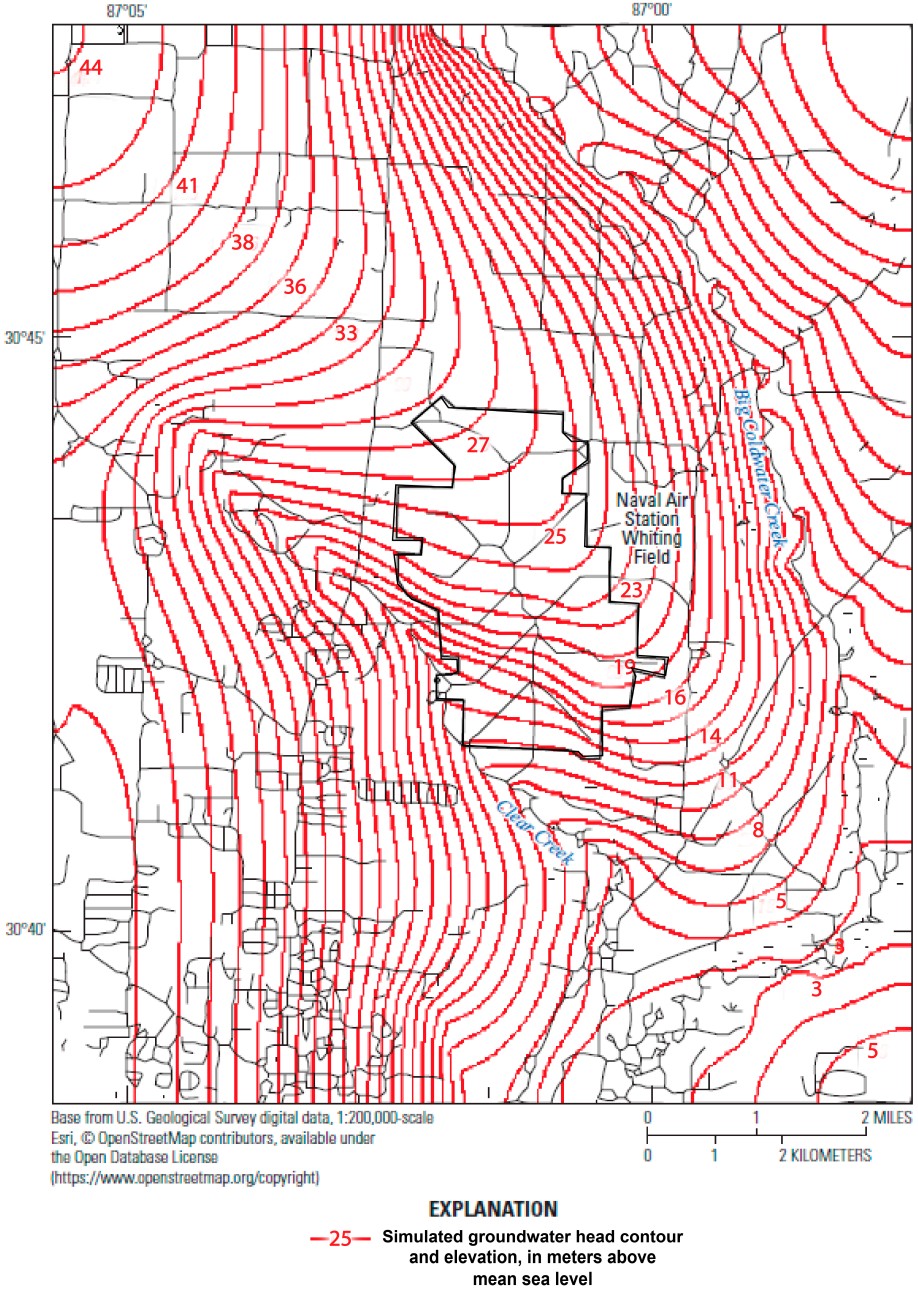

**Figure 7.** Vertically averaged simulated groundwater potentiometric contours in the Whiting Field groundwater model area, Naval Air Station Whiting Field, near Milton, Florida (from [3]).

This steady-state simulation had less variability in groundwater flows than a transient simulation, so a particle was backtracked from each well location at the depth of the screened interval and considered representative of that well (Figure 3). Multiple particles per well would produce little variations in the particle backtracking.

## 4. Results

The backtracking option in MODPATH was applied to the six well locations in NASWF. The paths of one particle per well were backtracked to the location of surficial recharge. As stated previously, three of the six wells have independently known travel time estimates based on previous groundwater quality sampling, which were compared to the particle-tracking estimations. Given that all six wells were screened for sand and gravel aquifer, the age date results were comparable for the three wells without known ages.

### 4.1. Particle Backtracking Locations

The backtracking of particles to their surficial recharge points for the six wells in this study, using the steady-state WFGM simulation, is shown in Figure 8A,B. The difference in particle paths between the without pumping (A) and with pumping (B) scenarios was more obvious near the pumping public supply wells within the NASWF boundary, W-N4, W-W3, and W-S2. The particle paths to these three wells were significantly longer in the pumping scenario (Figure 8B), reaching outside the NASWF boundary, whereas, in the no pumping scenario, the recharge locations were all within NASWF. For the monitoring wells, the particle path for well WHF-16-MW-7D was farther west in the pumping scenario, indicating that some of the water received by WHF-16-MW-7D in the no pumping scenario was taken up by pumping in wells W-N4, W-W3, and W-S2.

**A**

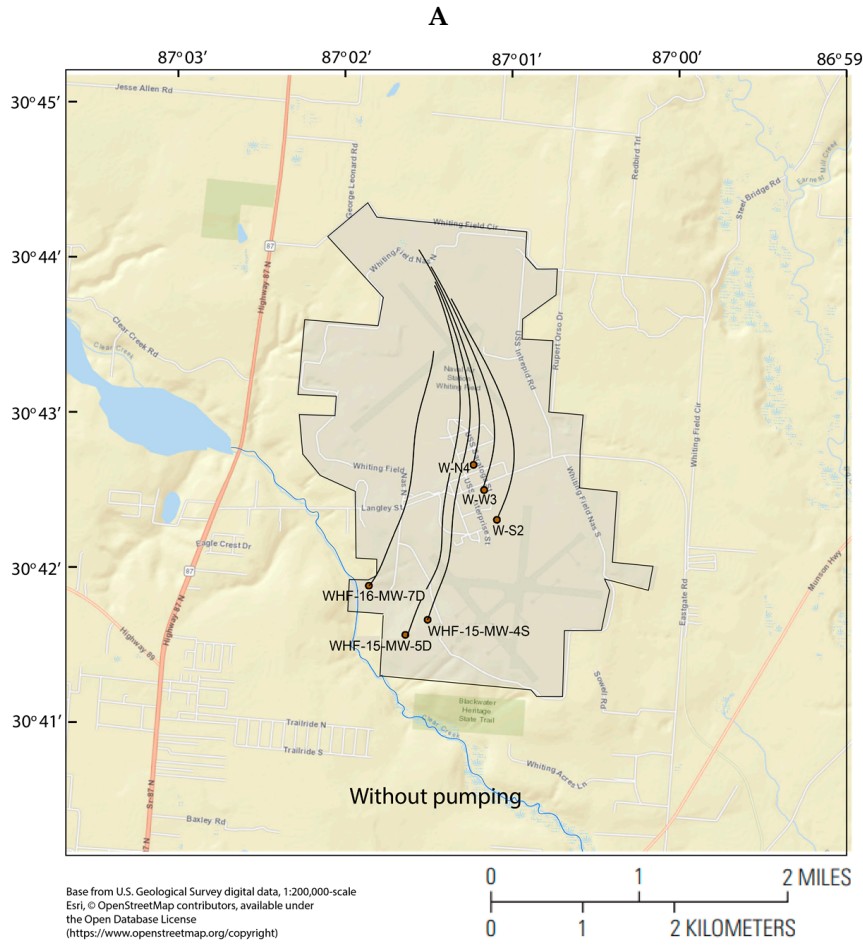

**Figure 8.** *Cont.*

**B**

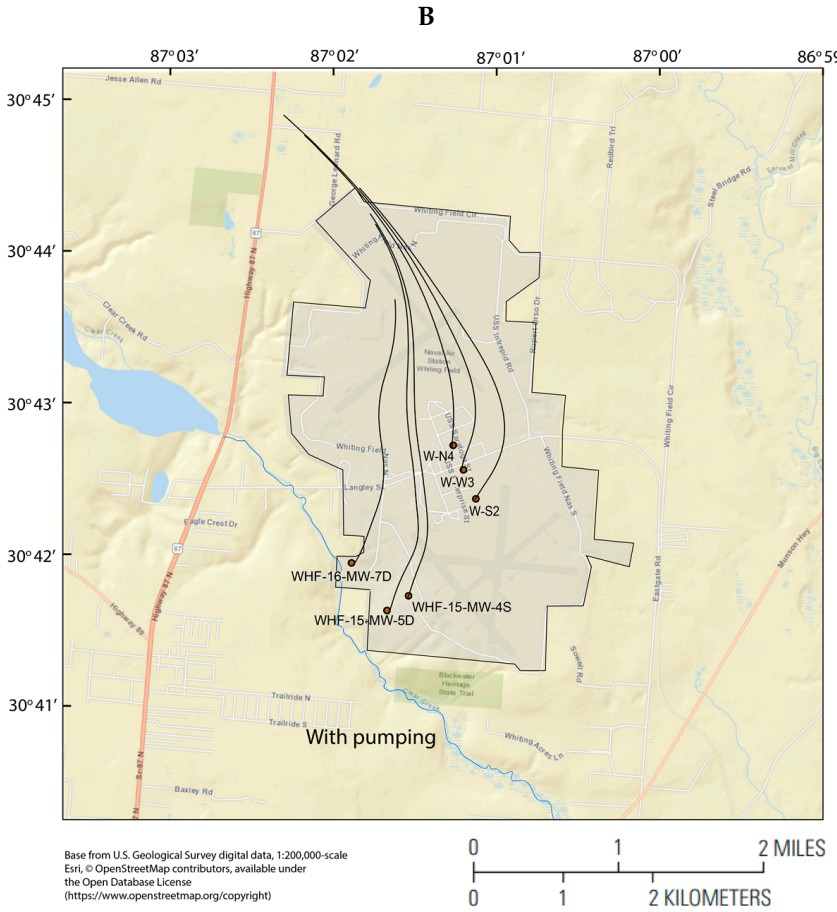

**Figure 8.** Locations of monitoring [WHF-n] and public supply [W-n] wells used in the particle-tracking simulations, where n is the specific well name. Particle backtracking results from all six wells (**A**) without and (**B**) with pumping are shown as single black lines that start at each well and travel upgradient to simulated recharge areas.

### 4.2. Particle Backtracking Locations

The time for particles to travel from the top of the water table, where water arrives as recharge from the unsaturated zone to each well, is listed in Table 2. The groundwater model was a steady-state simulation, and the well pumping rates were averaged over the period of the well's existence (Table 1). The particle tracking estimated travel times in the range of 26 to 46 years since recharge [3]. Pumping shortened the groundwater flow pathway travel time in one of the six wells, as might be assumed with higher groundwater velocities. The other five wells, including the three pumping wells within NASWF (W-N4, W-W3, and W-S2), drew groundwater from substantially farther away when they were pumping (Figure 8), hence the longer travel time. The other two wells with longer travel time when pumping than without pumping were WHF-15-MW-4S and WHF-15-MW-5D, whose flow paths were farther west when the three pumping wells within NASWF were pumping. The particle paths to these two wells were, therefore, longer when pumping was occurring (Figure 8).

Simulated particle travel times were similar to estimates from field data at the three monitoring wells where tritium and helium or CFC-11 concentrations were used to date the groundwater, with differences varying from 3.5 percent to 14.6 percent for the well pumping scenario (Table 2). This indicated that the hydraulic conductivity of the sand and gravel aquifer represented in the WFGM was reasonably accurate. It is interesting to note that at the monitoring well that was tritium- and helium-dated (WHF-16-MW-7D), the WFGM slightly overestimated the travel time and, at the two wells that were CFC-11-dated, WHF-15-MW-4S and WHF-15-MW-5D, the WFGM underestimated the travel time. This

was too small of a data set to make a definitive conclusion. The three pumping wells within NASWF were installed between 1947 and 1952, so the field estimates of travel time might be expected to correspond more closely to the WFGM pumping scenario. This was the case for all three wells where field data estimates were made (Table 2).

General comparisons can be made with the other studies mentioned in the Introduction section. The groundwater model in [9] was calibrated with the particle-tracking results; the WFGM could be recalibrated likewise, but adjusting porosity values would have yielded very predictable results (see Section 3 in Simulation Limitations). Solute transport simulations were compared with particle-tracking results in [10]. Solute transport simulations do yield more information than particle tracking and relate more to the quantities of interest (contaminant concentrations), but transport computations require far more information and computational effort than particle tracking and cannot be performed in reverse to locate sources. Forward particle tracking was used in [12] to determine travel times from recharge sites, but the study did not have isotope age-dating data to compare with. In order to find the source for a given well, this forward tracking must be performed iteratively by trial and error, unlike the backwards tracking in our study here.

**Table 2.** Simulated particle track travel times to each of the six wells.

| | Simulated Particle Travel Time, Years | | |
|---|---|---|---|
| **Well Name** | **No Pumping** | **Pumping** | **Estimated from Field Data [3]** |
| WHF-16-MW-7D | 26.3 | 25.9 | 25 |
| WHF-15-MW-4S | 31.3 | 33.3 | 39 |
| WHF-15-MW-5D | 31.4 | 33.4 | 38 |
| W-N4 | 29.4 | 37.6 | - |
| W-W3 | 31.2 | 43.7 | - |
| W-S2 | 35.8 | 45.9 | - |

## 5. Implications for Naval Air Station Whiting Field

The MODPATH particle-tracking simulation provided pathlines from selected wells to recharge sources. If the source areas identified by the model are accurate, several implications can be made:

1. The higher pumping rate at the three public supply wells lengthens simulated groundwater flow pathways to these wells and changes flow patterns nearby, as can be seen in the flow paths to wells WHF-16-MW-7D, WHF-15-MW-4S, and WHF-15-MW-5D (Figure 8).

2. Relating the simulated groundwater flow pathways to features in NASWF yields estimates of known and unknown source areas (Figure 9). Monitoring wells WHF-15-MW-4S and WHF-15-MW-5D have flow pathways that pass close to known landfill/disposal areas and industrial sites near the center of NASWF [3]. The flow pathway to WHF-16-MW-7D also passes a nearby landfill/disposal area and has a source near a firefighter training area in the north of NASWF (Figure 8). It was stated by [3] that the benzene contamination in WHF-16-MW-7D likely came from the underground storage tank site near the center of NASWF (Figure 9), but the particle-tracking results indicate that the flow path to WHF-16-MW-7D is not as close to the underground storage tanks as other wells. However, a possible explanation for this difference is that dispersion was not represented in the particle-tracking simulation and could allow for the additional movement of contaminants. Also, a source farther north might be possible.

3. The simulated sources of water for the three pumping public supply wells were farther north than those of the monitoring wells, and it seems that any contamination detected in monitoring wells WHF-15-MW-4S and WHF-15-MW-5D might also be detected in the three public supply wells. Moreover, the simulated groundwater flow pathways for the three public supply wells did not approach known firefighter

training areas (Figure 9). As such, at this time, the source of chemicals such as PFOS and PFOA detected in these wells remains a focus of future investigation. In fact, there may be sources off base, as shown by the flow pathways (Figure 9), in areas that are characterized by agricultural land uses.

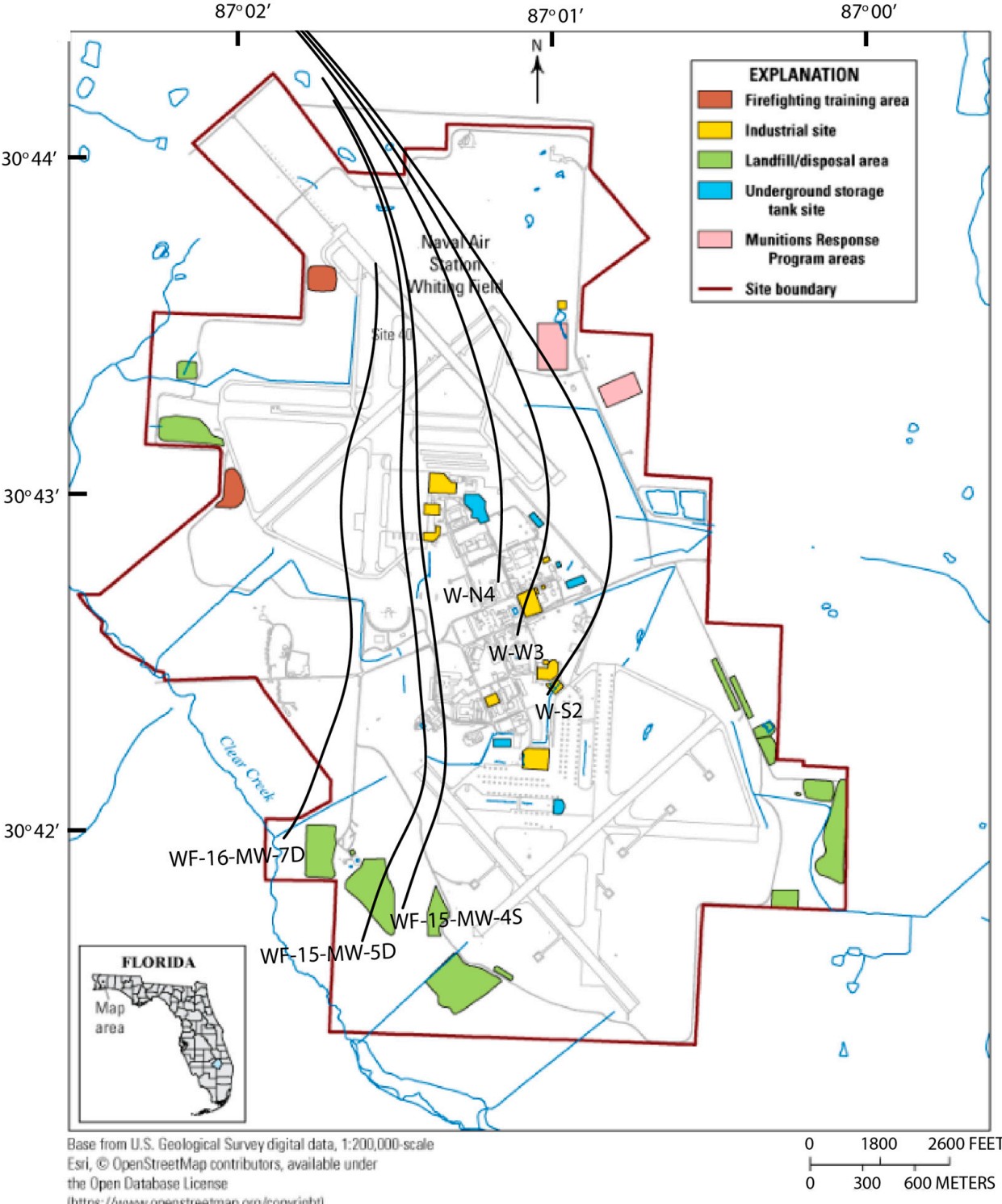

**Figure 9.** Known or suspected potential source areas of contamination within Naval Air Station Whiting Field and superimposed simulated groundwater flow pathways from the three monitoring wells and three public supply wells (modified from [3]).

## 6. Simulation Limitations

In order to use any groundwater flow model simulation properly, the capabilities and limitations of the simulation must be considered. The limitations of the WFGM are described in [3] and primarily involve the discretization of the model and limitations in available data. These limitations carry over to the MODPATH particle-tracking application. In addition, limitations in the MODPATH application itself include the following:

1.  MODPATH calculates the average linear velocity from the flow across a model cell face. Therefore, MODPATH cannot simulate the actual velocity distribution that occurs due to variations in the aquifer that cause dispersion. The sporadic nature of rainfall–recharge causes vertical groundwater gradients to temporally fluctuate, causing dispersion not represented in the simulation.

2.  The depth of the point where a particle reaches a well is approximated in the simulation by the depth of the well, but actual interchange between the aquifer and a well occurs over the well's screened interval. This discrepancy is likely to affect travel-time estimates as uncertainty in depth from surface recharge produces uncertainty in vertical transport times. However, particle pathlines are less likely to be affected as vertical variations in horizontal groundwater flow velocity and direction are minimal [3].

3.  Porosity, which inversely affects groundwater flow velocity, was simulated as spatially uniform and given a nominal value of 0.3. If there were additional field data, the porosity could be made spatially variable and calibrated with the measured travel time data. However, it was of interest to see how groundwater flow generated by the WFGM produced simulated particle travel time without specific calibration to the field data. With a steady-state groundwater simulation, porosity data are not used in the groundwater simulation and only act as a divisor for computing particle-tracking velocity from the computed flow. So, any spatially uniform change in porosity produces the same particle paths and changes the travel time by the ratio of the old porosity to the new porosity.

4.  The use of the steady-state WFGM simulation ignores possible seasonal and long-term fluctuations in groundwater flow magnitude and direction. Surficial recharge was calibrated in the WFGM for groundwater heads from August 1997, so the recharge and groundwater flows may not have reflected entire multi-decadal particle travel times. Seasonal fluctuations generally increase flow in the wet season and decrease flow in the dry season, with the steady-state simulation representing intermediate flows, a good long-term average.

5.  The groundwater flow velocities used by the particle-tracking application are interpolated between WFGM grid cells and cannot take any smaller-scale features into account. This limitation comes from the groundwater model simulation.

The limitations and uncertainty of the numerical groundwater model are more substantial than the limitations of the particle-tracking application. Given the calibration and comparison of the WFGM to field data [3], the groundwater model can be considered a reasonable representation of the system, and the similarity of particle travel times to field estimates indicates that the MODPATH particle-tracking application is useful and can make reasonable estimates of source locations and travel times.

## 7. Conclusions

The MODPATH particle-tracking application for the WFGM has given substantial insight into contaminant contribution areas in NASWF and how particle-tracking techniques can be used with field data at a site with aquifer contamination. The WFGM was developed for the area around NASWF, where past waste disposal has led to the contamination of the unsaturated zone and the development of plumes in the underlying sand and gravel aquifer. Model calibration efforts utilized lithologic logs to estimate hydraulic conductivity distribution, 59 groundwater monitoring wells to compare measured and simulated heads, and discrete flow measurements made at three surface water locations. The wells in NASWF chosen for particle-tracking analysis included three water supply pumping

wells within NASWF and three monitoring wells that had field estimates of travel time from the land surface to each well. These isotope-based travel time estimates allowed for a direct comparison with the particle-tracking results. The MODPATH simulation was performed with the steady-state version of the WFGM, and simulations with and without well pumping were performed.

The simulated times of travel matched those at the three public supply wells, with differences from 3.5 percent to 14.6 percent for the well pumping scenario, and the difference in travel times between the no pumping and pumping scenarios were greatest at those wells with the most pumping. The source areas for monitoring wells, identified by the particle tracking, include areas within NASWF that contain known or suspected contamination sources, supporting the findings of past field groundwater sampling. Based on the particle paths in the simulation with well pumping, potential sources of pollution could be outside NASWF, especially for the three public supply wells. The possibility that sources outside NASWF might contribute to these wells was not seriously considered before these simulation results, but these credible results indicate a larger contribution area. It must be noted that known or suspected sources of contamination may lie outside of direct particle paths, so assumptions may be made as to dispersion around particle paths, which was not represented in the particle tracking. The similarity of the simulated travel times and those estimated from field samples indicates that the WFGM reasonably represents groundwater velocities and flow directions, providing an additional model check. This demonstrates the utility of combining isotope data with particle-tracking simulation for a more complete conceptual model.

**Author Contributions:** Conceptualization, E.D.S. and J.E.L.; methodology, E.D.S.; software, E.D.S.; validation, E.D.S. and J.E.L.; formal analysis, E.D.S.; investigation, J.E.L. and E.D.S.; resources, J.E.L., M.A.S. and S.E.P.; data curation, E.D.S.; writing—original draft preparation, E.D.S. and J.E.L.; writing—review and editing, M.A.S. and S.E.P.; visualization, E.D.S.; supervision, J.E.L.; project administration, J.E.L.; funding acquisition, J.E.L. All authors have read and agreed to the published version of the manuscript.

**Funding:** This research was funded by the U.S. Navy Naval Facilities Engineering Systems Command Southeast.

**Data Availability Statement:** The computer model code, input files, and output files can be accessed at https://doi.org/10.5066/P9M0OD8F.

**Acknowledgments:** The authors thank the following individuals for contributing to the study: Brooke Boyd, John "Jeff" Kissler, Jonathan Stewart, Billy Ryan, and Charles Egri of the Installation Environmental Program, Naval Air Station Whiting Field; Alex Eddington, Bill Duffy, and Ryan Samuels of Resolution Consultants, Inc.; Sam Naik, of Tetra Tech; and the journal reviewers at Hydrology. Any use of trade, firm, or product names is for descriptive purposes only and does not imply endorsement by the U.S. Government.

**Conflicts of Interest:** The authors declare no conflicts of interest. The funders had no role in the design of the study; in the collection, analyses, or interpretation of data; in the writing of the manuscript; or in the decision to publish the results.

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
