# Peer review of "Insight into Sources of Benzene, TCE, and PFOA/PFOS in Groundwater at Naval Air Station Whiting Field, Florida, through Numerical Particle-Tracking Simulations"

_hydrology, doi:10.3390/hydrology11030037_

Round 1

Reviewer 1 Report

Comments and Suggestions for Authors

Review of the article Insight into transport of benzene, TCE, and PFAS in groundwater at Naval Air Station Whiting Field, Florida, through numerical particle tracking simulations.

The paper is more a case study or report and not an article, given that the announced goal is ''the use of a particle-tracking algorithm in the groundwater model of NASWF to estimate sources of contaminants detected in monitoring wells and public-supply wells at NASWF.''

The Introduction must contain the state of the art in the field, which is missing.

Most of the information in the Introduction must be moved to the second section.

The novelty of the research must be emphasized in the Introduction. From the actual presentation, the reader cannot catch the novelty after reading the first section. 

Many figures must be better explained.

The research is not reproducible at this moment based on the provided information. 

Comparison with the results obtained by other scientists using the same software must be done.

Conclusions must replace the summary.

Overall, the work is not at the level required for the publication in Hydrology.

Author Response

Review 1

Open Review

Quality of English Language

( ) I am not qualified to assess the quality of English in this paper
( ) English very difficult to understand/incomprehensible
( ) Extensive editing of English language required
( ) Moderate editing of English language required
( ) Minor editing of English language required
(x) English language fine. No issues detected

Yes

Can be improved

Must be improved

Not applicable

Does the introduction provide sufficient background and include all relevant references?

( )

( )

(x)

( )

Are all the cited references relevant to the research?

( )

( )

(x)

( )

Is the research design appropriate?

( )

( )

(x)

( )

Are the methods adequately described?

( )

( )

(x)

( )

Are the results clearly presented?

( )

( )

(x)

( )

Are the conclusions supported by the results?

( )

( )

(x)

( )

Comments and Suggestions for Authors

Review of the article Insight into transport of benzene, TCE, and PFAS in groundwater at Naval Air Station Whiting Field, Florida, through numerical particle tracking simulations.

The paper is more a case study or report and not an article, given that the announced goal is ''the use of a particle-tracking algorithm in the groundwater model of NASWF to estimate sources of contaminants detected in monitoring wells and public-supply wells at NASWF.''

We thank the reviewer for this comment. To reconcile it, we have revised the manuscript to include new sentences that point out that although the characteristics of the study site are specific to the study area, the contaminants encountered in groundwater, such as benzene and TCE, are two of the most commonly detected in groundwater at thousands of sites around the world. And the detection of PFAS/PFOA is becoming increasingly so. As such, the work we present in the revised paper has strong transferability to the widespread audience of Hydrology Journal. Moreover, even prior to this change, we would like to point out that another peer reviewer stated that “This work may be of potential interest to a wide range of readers of the Hydrology journal”.

The Introduction must contain the state of the art in the field, which is missing.

We are in agreement with this review comment. Substantial background information has been added to the revised Introduction, including “Currently, particle tracking algorithms have been developed for unstructured model grids, limited to smoothed, rectangular-based quadtree and quadpatch grids (Pollack, 2016). Another particle tracking algorithm has been developed for MODFLOW for unstructured grids and has been demonstrated for cases where jnterpolation between model cells is difficult due to high spatial variability in groundwater flow (Craig et al, 2019). Particle-tracking algorithms have been developed for a finite-element groundwater model in two dimensions that takes into account the interpolated velocities within the model elements (Cayar et al. 2010), This was compared with MODPATH which uses the finite-difference MODFLOW groundwater model.” To sum, we have addressed this comment by the inclusion of three new citations with references.

Most of the information in the Introduction must be moved to the second section.

We concur. The bottom part of the first paragraph of the Introduction, describing details about the contaminants in the NASWF, has been moved to the end of the Study Area section.

The novelty of the research must be emphasized in the Introduction. From the actual presentation, the reader cannot catch the novelty after reading the first section. 

We appreciate the insight provided by this comment. We hopefully have rectified the issue in the following revision, as we have added the following text to the Introduction: “The particle-tracking simulation for the WFGM has a special advantage in isotope age-dating data collected at wells in the study area. The comparison of these allows for an evaluation of the travel times and supports the modeled travel distances. The particle-tracking results lead to a reevaluation and expansion of the area considered as potentially contributing to contamination at test well locations. Previous to the particle-tracking simulation, only sources within Whiting Field were considered. This indicates the particle-tracking technique applied can delineate contribution areas more quantitatively and help determine approaches to contaminant management.”

Many figures must be better explained.

Scale and geographic coordinates added to figures 1, 2, 5, and 6. Figures 3 and 4 are improved and details accentuated.

The research is not reproducible at this moment based on the provided information. 

Thank you for the comment. We would like to point out to the reviewer that the groundwater model is completely in an online archive [6] that is downloadable. This will be highlighted in this article along with the necessary information for a user to include particle data in MODPATH. As such, anyone can run the model and reproduce the results.

Comparison with the results obtained by other scientists using the same software must be done.

We concur with the comment and several applications of particle tracking to sites have been added to the Introduction (Cousquer et al, 2018; Yidana, 2011; Gusyev et al, 2014; Jing et al, 2021) with discussion of how these studies relate to this article.

Conclusions must replace the summary.

The summary has been changed to Conclusions and the points described in the previous discussion of the novelty of the research are expanded upon.

Overall, the work is not at the level required for the publication in Hydrology.

We have to disagree with the reviewer on this particular comment. We would like to make note that, as stated in response to a previous comment, another reviewer commented the exact opposite conclusion, that “This work may be of potential interest to a wide range of readers of the Hydrology journal”. Consequently, we feel that the changes discussed in the responses above should satisfy the requirements for Hydrology Journal.

Reviewer 2 Report

Comments and Suggestions for Authors

It could be explained or discussed in more detail how the porosity considered in your model validates the data found for flow velocity, simulations were carried out with values different from the 0.3 mentioned and the historical data can be used in this section of your model.

The aspect of steady state simulation that does not consider seasonal fluctuations which range of error or variability may represent in your project described

When particle trajectories are mentioned in the simulation with well pumping, it is argued that the sources of contamination could be outside, this aspect can be described or explained in more detail.

Author Response

Review 2

Open Review

Quality of English Language

(x) I am not qualified to assess the quality of English in this paper
( ) English very difficult to understand/incomprehensible
( ) Extensive editing of English language required
( ) Moderate editing of English language required
( ) Minor editing of English language required
( ) English language fine. No issues detected

Yes

Can be improved

Must be improved

Not applicable

Does the introduction provide sufficient background and include all relevant references?

( )

(x)

( )

( )

Are all the cited references relevant to the research?

( )

(x)

( )

( )

Is the research design appropriate?

(x)

( )

( )

( )

Are the methods adequately described?

(x)

( )

( )

( )

Are the results clearly presented?

( )

(x)

( )

( )

Are the conclusions supported by the results?

( )

(x)

( )

( )

Comments and Suggestions for Authors

It could be explained or discussed in more detail how the porosity considered in your model validates the data found for flow velocity, simulations were carried out with values different from the 0.3 mentioned and the historical data can be used in this section of your model.

We thank the reviewer for this comment. With a steady-state groundwater simulation, the porosity data are not used in the groundwater simulation and only act as a divisor for computing particle-tracking velocity from computed flow. So, any spatially uniform change in porosity produces the same particle paths and changes the travel time by the ratio of the old porosity to the new porosity. This process has been explained as an addition to item 3 in the Simulation Limitations section.

The aspect of steady state simulation that does not consider seasonal fluctuations which range of error or variability may represent in your project described

The error induced by using a steady-state simulation instead of a transient simulation has not been evaluated for this application, but one would expect the long-term pathlines to be very similar, but seasonal movement would be more variable. This potential difference will be elaborated in further detail in part 4 of Simulation Limitations.

When particle trajectories are mentioned in the simulation with well pumping, it is argued that the sources of contamination could be outside, this aspect can be described or explained in more detail.

Yes, we are in agreement with this point, and discussion of the simulations showing that sources can be outside Naval Station Whiting Field, not considered before, is significant and was included in the original manuscript. However, this point will be added to the Conclusion section.

Submission Date

31 October 2023

Date of this review

15 Nov 2023 15:31:12

Bottom of Form

© 1996-2023 MDPI (Basel, Switzerland) unless otherwise stated

Reviewer 3 Report

Comments and Suggestions for Authors

General comments

The manuscript  titled “Insight into transport of benzene, TCE, and PFAS in groundwater at Naval Air Station Whiting Field, Florida, through numerical particle tracking simulations” raises an interesting topic and concerns an interesting area of research. This work may be of potential interest to a wide range of readers of the Hydrology journal. The manuscript is interestingly written and I read it with interest. However, at this stage it requires far-reaching changes. Numerous fragments of the manuscript require more extensive commentary and/or supplementation. The manuscript may also be shortened in some parts.

At this stage it is unclear what the appropriate nature of this work is. Authors should clearly indicate what their manuscript brings new to the research performed by Landmeyer et al., 2021 (doi.org/10.3133/sir20215124). Is the novelty of this work the determination of the trajectory of pollutant particles migrating in the aquifer? This is undoubtedly an important issue, but it is a strictly local problem. What significance do these observations have for a reader from, for example, Asia or the EU? Therefore, at the current stage, the novelty of this manuscript is difficult to grasp. References to the contemporary state of knowledge regarding the issues discussed are also insufficient.

The discussion of results section is, in my opinion, the weakest part of this manuscript. Generally, there are no references to tracer and model studies in hydrogeology. There is also no broader discussion of the results obtained by the Authors in the context of other authors' research carried out using similar research tools (digital models and gaseous environmental tracers, i.e. CFCs, helium, etc.).

Detailed comments are provided below:

Lines 68-71: The information from these lines is basically repeated again in lines 107-108.

Lines 55 and 132: Figures 1 and 2 can be merged without losing the quality of the information presented.

Line 85 and further: “…from 150 to 190 ft..” – please use SI units. This applies to the entire manuscript.

Lines 93-98: The description of hydrogeological conditions is too simplified. At least one hydrogeological cross-section and a map of the groundwater table would be useful to better understand the problem.

Lines 116-118: The dating of CFC-11 requires a bit wider explanation. Generally, CFC-11 is stable under oxidizing conditions. The authors indicate the existence of a transitional/oxidizing environment, but does this apply to the entire research area? Another issue is the methodology of measuring oxygen dissolved in groundwater. Was a flow cell used to enable measurements without access to atmospheric air? A scientific article should be fully clear and readable without the need to refer to external sources.

Line 120: "... Tritium (from helium ingrowth) concentrations ..." - this statement is unclear and, in my opinion, requires a broader comment from the authors. What source of tritium are the Authors mentioning here?

Line 180 and 197: Figures 3 and 4 are almost illegible, maybe it's worth cropping the unnecessary background? Moreover, as before, it is worth considering merging Figures 3 and 4 because they present very similar information.

Line 161: "... with the aquifer hydraulic conductivity" – OK, but what are the specific filtration parameters of the analyzed aquifer system?

Line 166-171: ".. . evapotranspiration extinction depths ..." OK. Underground evaporation generally depends on the climate, soil type and type of aquifer, but is usually is significantly up to a few meters from the ground surface. So the question arises - how much part of the WFGM area can this phenomenon affect?

Lines 218-237: In my opinion, this description is too broad. The manuscript without loss of quality and can be shortened by this fragment and you can simply provide appropriate references.

Line 261: Since this is a results section, it should be clearly indicated what is the source (reference) of the groundwater age results discussed by the Authors.

Lines 273 and further: Dating such young waters using the helium method may be subject to an error of an amount that is difficult to determine, which results, among other things, from the uncertainty of determining the size of the helium flux (external) from the deeper bedrock. Next issue is CFC-11 used for dating is susceptible to environmental conditions and may decompose with low oxygen content and the participation of microorganisms (microbial breakdown). (see more, Kotowski 2019, 2023 and many other works on this topic). Generally, there is no broader discussion regarding the uncertainty of groundwater dating, and the results of this dating are important for the considerations conducted by the Authors in this work. Consequently, the question arises about the quality of the results obtained in this way. Simulation models are not magical black boxes and the quality of the simulation depends crucially on the quantity/quality of the modeling input data. Regardless of the mathematics used, it will be an approximate description and, moreover, it will depend on the quantity and quality of the input measurement/laboratory data, and not on the data obtained using any digital model.

Lines 317 and further: The discussion on the limitations of numerical modeling concerns issues that are known and are quite general in nature. In my opinion, this paragraph could be shortened and this thread could be expanded to include a discussion of the uncertainty related to the dating of groundwater (comment to line 273) or the differentiation of the values of hydraulic parameters used in modeling. For example, the Authors assume a porosity value of 0.3 for the entire research area. At the same time, they indicate that, "...it was of interest to see how groundwater flow generated by the WFGM produced simulated particle travel time without specific calibration to the field data. " APPROX. , but in my opinion it would be even more interesting to compare the modeling carried out for real (field) and simplified values (only 0.3). This would allow, among other things, to answer the question whether this is not too far-reaching a simplification. The issue of possible differences in the calibration of such models would also be interesting.

Lines 355 and further: Summary ?? Shouldn't it be a "Conclusion" chapter presenting the most important conclusions resulting from the research conducted by the authors? In its current form, the summary chapter actually conveys content similar to that presented in the abstract. Since I believe that changes in the text should also include the summary chapter, I do not refer to this part of the manuscript now.

Example references:

Kotowski T., Najman J., Nowobilska-Luberda A., Bergel T. and Kaczor G., 2023. Analysis of the interaction between surface water and groundwater using gaseous tracers in a dynamic test at a riverbank filtration intake. Hydrological Processes, 37(4),  e14862, doi.org/10.1002/hyp.14862

Kotowski T., Chudzik L., Najman J., 2019 - Application of dissolved gases concentration measurements, hydrochemical and isotopic data to determine the circulation conditions and age of groundwater in the Central Sudetes Mts. Journal of Hydrology, 569: 735-752. (doi.org/10.1016/j.jhydrol.2018.12.013)

Author Response

Review 3

Open Review

(x) I would not like to sign my review report

( ) I would like to sign my review report

Quality of English Language

(x) I am not qualified to assess the quality of English in this paper

( ) English very difficult to understand/incomprehensible

( ) Extensive editing of English language required

( ) Moderate editing of English language required

( ) Minor editing of English language required

( ) English language fine. No issues detected

Yes      Can be improved          Must be improved         Not applicable

Does the introduction provide sufficient background and include all relevant references?

( )         (x)        ( )         ( )

Are all the cited references relevant to the research?

( )         ( )         (x)        ( )

Is the research design appropriate?

( )         (x)        ( )         ( )

Are the methods adequately described?

( )         (x)        ( )         ( )

Are the results clearly presented?

( )         ( )         (x)        ( )

Are the conclusions supported by the results?

( )         ( )         (x)        ( )

Comments and Suggestions for Authors

General comments

The manuscript  titled “Insight into transport of benzene, TCE, and PFAS in groundwater at Naval Air Station Whiting Field, Florida, through numerical particle tracking simulations” raises an interesting topic and concerns an interesting area of research. This work may be of potential interest to a wide range of readers of the Hydrology journal. The manuscript is interestingly written and I read it with interest. However, at this stage it requires far-reaching changes. Numerous fragments of the manuscript require more extensive commentary and/or supplementation. The manuscript may also be shortened in some parts.

We would like to thank this reviewer for pointing out that This work may be of potential interest to a wide range of readers of the Hydrology journal.” The manuscript has been revised to make the case that this research approach and results have application to similar contamination sites around the world.

At this stage it is unclear what the appropriate nature of this work is. Authors should clearly indicate what their manuscript brings new to the research performed by Landmeyer et al., 2021 (doi.org/10.3133/sir20215124). Is the novelty of this work the determination of the trajectory of pollutant particles migrating in the aquifer? This is undoubtedly an important issue, but it is a strictly local problem. What significance do these observations have for a reader from, for example, Asia or the EU? Therefore, at the current stage, the novelty of this manuscript is difficult to grasp. References to the contemporary state of knowledge regarding the issues discussed are also insufficient.

We appreciate this comment and the opportunity to provide clarifying comments. Although determining contaminant transport at Whiting Field is a local problem (or national problem as it is a Superfund site), the significance of this study to the greater scientific community is addressed by the following text that has been added to the revised Introduction: “The particle-tracking simulation for the WFGM has a special advantage in isotope age-dating data collected at wells in the study area. The comparison of these allows for an evaluation of the travel times and supports the modeled travel distances. The particle-tracking results lead to a reevaluation and expansion of the area considered as potentially contributing to contamination at test well locations. Previous to the particle-tracking simulation, only sources within Whiting Field were considered. This indicates the particle-tracking technique applied can delineate contribution areas more quantitatively and help determine approaches to contaminant management.”

The discussion of results section is, in my opinion, the weakest part of this manuscript. Generally, there are no references to tracer and model studies in hydrogeology. There is also no broader discussion of the results obtained by the Authors in the context of other authors' research carried out using similar research tools (digital models and gaseous environmental tracers, i.e. CFCs, helium, etc.).

Thank you for this comment. TO reconcile, we added reference to associated research, such as (Cousquer et al, 2018; Yidana, 2011; Gusyev et al, 2014; Jing et al, 2021) which show other applications of particle-tracking techniques. Brief comparisons to these studies have been included at the beginning of the revised Results section.

Detailed comments are provided below:

Lines 68-71: The information from these lines is basically repeated again in lines 107-108.

The second reference is removed.

Lines 55 and 132: Figures 1 and 2 can be merged without losing the quality of the information presented.

Figure 1 is to a scale showing a larger area, helping in site location. Figure 2 shows more detail. So we would like to keep them separate instead of having one map to a single scale.

Line 85 and further: “…from 150 to 190 ft..” – please use SI units. This applies to the entire manuscript.

Units have been changed to SI units in entire manuscript.

Lines 93-98: The description of hydrogeological conditions is too simplified. At least one hydrogeological cross-section and a map of the groundwater table would be useful to better understand the problem.

A figure showing the stratigraphic column is added with an explanation of the layers. There is limited information on the spatial distribution of the water table so figure 5 is already the best representation.

Lines 116-118: The dating of CFC-11 requires a bit wider explanation. Generally, CFC-11 is stable under oxidizing conditions. The authors indicate the existence of a transitional/oxidizing environment, but does this apply to the entire research area? Another issue is the methodology of measuring oxygen dissolved in groundwater. Was a flow cell used to enable measurements without access to atmospheric air? A scientific article should be fully clear and readable without the need to refer to external sources.

All wells had measurable DO. Yes, a flow cell was used. We have added this requested information to the paper, as: “Before sample collection, groundwater was pumped through a low-flow chamber and measurements of physical properties and chemical constituents of groundwater, such as dissolved oxygen, pH, specific conductance, and temperature, were measured using a YSI 6920 sonde (YSI, Inc.). The sonde was calibrated daily before sampling using appropriate standard methods for dissolved oxygen, pH, and specific conductance as reported in the USGS National Field Manual (U.S. Geological Survey, variously dated). Groundwater samples were collected after measurements of dissolved oxygen, pH, specific conductance, and temperature had stabilized. Groundwater did not require filtration because of the low sample turbidity”.

Line 120: "... Tritium (from helium ingrowth) concentrations ..." - this statement is unclear and, in my opinion, requires a broader comment from the authors. What source of tritium are the Authors mentioning here?

We understand that this discussion could have been made clearer. We have revised the paper to include comments that we are measuring the tritium from helium ingrowth, as 3H decays to 3He. As such, tritium can once again become a useful indicator of age, even though most of the “bomb peak” tritium has now reached background levels.

Line 180 and 197: Figures 3 and 4 are almost illegible, maybe it's worth cropping the unnecessary background? Moreover, as before, it is worth considering merging Figures 3 and 4 because they present very similar information.

Surface details and numbers are accentuated for clarity in these figures. The numbering for the lithologic logs was unnecessary and crowded the figure, so they were removed. With so many points for lithologic cores and monitoring wells, we believe that combining the two figures would create too much crowding and confusion.

Line 161: "... with the aquifer hydraulic conductivity" – OK, but what are the specific filtration parameters of the analyzed aquifer system?

The lithologic logs were not implicitly tested for hydraulic conductivity as they were not collected with model development in mind. They were used to proportion the model hydraulic conductivities that had been calibrated.

Line 166-171: ".. . evapotranspiration extinction depths ..." OK. Underground evaporation generally depends on the climate, soil type and type of aquifer, but is usually is significantly up to a few meters from the ground surface. So the question arises - how much part of the WFGM area can this phenomenon affect?

It is assumed that there is some significant evaporation directly from the saturated aquifer in areas with shallow groundwater, which is most common in the southern part of the study area and near the streams. This is likely smaller than evaporation from the unsaturated zone and transpiration. In the model, these quantities are lumped together and, along with recharge from precipitation calibrated within the model.

Lines 218-237: In my opinion, this description is too broad. The manuscript without loss of quality and can be shortened by this fragment and you can simply provide appropriate references.

This section has been removed from the revised manuscript.

Line 261: Since this is a results section, it should be clearly indicated what is the source (reference) of the groundwater age results discussed by the Authors.

Reference [3] is added to the description.

Lines 273 and further: Dating such young waters using the helium method may be subject to an error of an amount that is difficult to determine, which results, among other things, from the uncertainty of determining the size of the helium flux (external) from the deeper bedrock. Next issue is CFC-11 used for dating is susceptible to environmental conditions and may decompose with low oxygen content and the participation of microorganisms (microbial breakdown). (see more, Kotowski 2019, 2023 and many other works on this topic). Generally, there is no broader discussion regarding the uncertainty of groundwater dating, and the results of this dating are important for the considerations conducted by the Authors in this work. Consequently, the question arises about the quality of the results obtained in this way. Simulation models are not magical black boxes and the quality of the simulation depends crucially on the quantity/quality of the modeling input data. Regardless of the mathematics used, it will be an approximate description and, moreover, it will depend on the quantity and quality of the input measurement/laboratory data, and not on the data obtained using any digital model.

The groundwater age dating results described in the current submission were discussed in Landmeyer and others (2022). The original manuscript included a citation to this reference. No change made.

Lines 317 and further: The discussion on the limitations of numerical modeling concerns issues that are known and are quite general in nature. In my opinion, this paragraph could be shortened and this thread could be expanded to include a discussion of the uncertainty related to the dating of groundwater (comment to line 273) or the differentiation of the values of hydraulic parameters used in modeling. For example, the Authors assume a porosity value of 0.3 for the entire research area. At the same time, they indicate that, "...it was of interest to see how groundwater flow generated by the WFGM produced simulated particle travel time without specific calibration to the field data. " APPROX. , but in my opinion it would be even more interesting to compare the modeling carried out for real (field) and simplified values (only 0.3). This would allow, among other things, to answer the question whether this is not too far-reaching a simplification. The issue of possible differences in the calibration of such models would also be interesting.

The field information on aquifer properties is insufficient to do anything but apply a multiplier to the porosity to make travel times match he age-dating. With a steady-state groundwater simulation, the porosity data is not used in the groundwater simulation and only acts as a divisor for computing particle-tracking velocity from computed flow. So any spatially uniform change in porosity produces the same particle paths and changes the travel time by the ratio of the old porosity to the new porosity. This is explained in detail with added text to the revised manuscript in Simulation Limitations part 3.

Lines 355 and further: Summary ?? Shouldn't it be a "Conclusion" chapter presenting the most important conclusions resulting from the research conducted by the authors? In its current form, the summary chapter actually conveys content similar to that presented in the abstract. Since I believe that changes in the text should also include the summary chapter, I do not refer to this part of the manuscript now.

The Summary is changed to Conclusions and the major results of the study, especially the comparison to age-dating and the model indicating that the source area can extend outside NASWF.

Example references:

Kotowski T., Najman J., Nowobilska-Luberda A., Bergel T. and Kaczor G., 2023. Analysis of the interaction between surface water and groundwater using gaseous tracers in a dynamic test at a riverbank filtration intake. Hydrological Processes, 37(4),  e14862, doi.org/10.1002/hyp.14862

Kotowski T., Chudzik L., Najman J., 2019 - Application of dissolved gases concentration measurements, hydrochemical and isotopic data to determine the circulation conditions and age of groundwater in the Central Sudetes Mts. Journal of Hydrology, 569: 735-752. (doi.org/10.1016/j.jhydrol.2018.12.013)

These references will be included in the discussion of age dating.

Submission Date

31 October 2023

Date of this review

10 Nov 2023 19:52:42

Reviewer 4 Report

Comments and Suggestions for Authors

This manuscript provides an insightful analysis of the impact of past waste-disposal activities at Naval Air Station Whiting Field (NASWF) on groundwater quality.  The utilization of the Whiting Field Groundwater Model (WFGM) and MODPATH particle-tracking application showcases an approach to understand groundwater-flow pathways, and potential source areas. The integration of field estimates, such as tritium and Trichlorofluoromethane concentrations, with particle-tracking simulations adds a valuable layer of validation to the calibrated WFGM, reinforcing the reliability of the groundwater-flow model. Additionally, the identification of source areas for monitoring wells through particle-tracking simulations aligns with water sampling findings, emphasizing the manuscript's contribution to addressing environmental concerns at NASWF. Below are minor review comments that may help the authors to improve the manuscript.

Ahmed S. Elshall

Review Comments:

Figure 1 and similar figures: Map has no scale or axis

L102 to improve solution of unconfined flow

L120-121: Unclear. Do you mean you that the date of Tritium sample that was taken was 1998? Please provide more information.

L123-125: How did you estimate porosity?

L146-160: Add a figure showing horizontal and vertical model grid, boundary conditions, and discretization of the drain.

L164: Provide the data source for precipitation data and information about your precipitation data

L165-167: Provide data source for GW table data or show you analysis steps

L167-171: How did you estimate ET? Estimating both recharge and hydraulic conductivity simultaneously with inverse modeling without data to constrain either of them can lead to an ill-posed model.

L173-197: Optional: You provided a source for your lithologic and hydraulic conductivity analysis. It might be useful to show additional details and show an example for this analysis to make the paper self-contained (e.g., Elshall et al., 2013; Elshall & Tsai, 2014).

L202-204: Before showing your simulated head, show first or provide a source for your calibration results (e.g., residuals vs, observed, simulated vs. observed, sum of squared residuals, etc.), estimate hydraulic conductivity field, estimated porosity, and the other model parameters that you calibrated.

L214 Figure 5: You might want to use different colors for four contours. For example, I cannot easily see the GW divide that you mentioned early with just one color?

L235: 237: What is number of particles and the initial particle locations at each well (e.g., Elshall et al., 2020). Please state if results are sensitive to the initial practical locations and number.

L261: “The time for particles to travel from the land surface” This is unclear. Do you mean from your water table? Are you also simulating the unsaturated zone?

L271: What is the start and end dates of your simulations or this is a steady-state simulation? You assumed a constant pumping rate throughout the simulation. Please confirm.

L281-284L How would you explain the similarity between particle travel time with and without pumping?

Elshall, A. S., Ye, M., & Finkel, M. (2020). Evaluating two multi-model simulation–optimization approaches for managing groundwater contaminant plumes. Journal of Hydrology, 590, 125427. https://doi.org/10.1016/j.jhydrol.2020.125427

Elshall, A.S., & Tsai, F. T.-C. (2014). Constructive epistemic modeling of groundwater flow with geological structure and boundary condition uncertainty under the Bayesian paradigm. Journal of Hydrology, 517. https://doi.org/10.1016/j.jhydrol.2014.05.027

Elshall, A.S., Tsai, F. T.-C., & Hanor, J. S. (2013). Indicator geostatistics for reconstructing Baton Rouge aquifer-fault hydrostratigraphy, Louisiana, USA. Hydrogeology Journal, 21(8). https://doi.org/10.1007/s10040-013-1037-5

Author Response

Review 4

Open Review

( ) I would not like to sign my review report

(x) I would like to sign my review report

Quality of English Language

( ) I am not qualified to assess the quality of English in this paper

( ) English very difficult to understand/incomprehensible

( ) Extensive editing of English language required

( ) Moderate editing of English language required

( ) Minor editing of English language required

(x) English language fine. No issues detected

Yes      Can be improved          Must be improved         Not applicable

Does the introduction provide sufficient background and include all relevant references?

(x)        ( )         ( )         ( )

Are all the cited references relevant to the research?

(x)        ( )         ( )         ( )

Is the research design appropriate?

(x)        ( )         ( )         ( )

Are the methods adequately described?

( )         (x)        ( )         ( )

Are the results clearly presented?

(x)        ( )         ( )         ( )

Are the conclusions supported by the results?

(x)        ( )         ( )         ( )

Comments and Suggestions for Authors

This manuscript provides an insightful analysis of the impact of past waste-disposal activities at Naval Air Station Whiting Field (NASWF) on groundwater quality.  The utilization of the Whiting Field Groundwater Model (WFGM) and MODPATH particle-tracking application showcases an approach to understand groundwater-flow pathways, and potential source areas. The integration of field estimates, such as tritium and Trichlorofluoromethane concentrations, with particle-tracking simulations adds a valuable layer of validation to the calibrated WFGM, reinforcing the reliability of the groundwater-flow model. Additionally, the identification of source areas for monitoring wells through particle-tracking simulations aligns with water sampling findings, emphasizing the manuscript's contribution to addressing environmental concerns at NASWF. Below are minor review comments that may help the authors to improve the manuscript.

Ahmed S. Elshall

Thank you for your review and we will address the review comments below.

Review Comments:

Figure 1 and similar figures: Map has no scale or axis

Scale and geographic coordinates have been added to revised figures 1, 2, 5, and 6.

L102 to improve solution of unconfined flow

The sentence has been changed to “The WFGM application uses the MODFLOW-NWT code, which uses a Newton formulation of MODFLOW-2005 [7] to improve computation of unconfined groundwater flow in a three-dimensional grid while representing hydrologic factors such as…”

L120-121: Unclear. Do you mean you that the date of Tritium sample that was taken was 1998? Please provide more information.

We appreciate the opportunity to respond to this comment. The revised manuscript has been changed to “A groundwater age of 1973 was estimated for this well using tritium (3H) and its daughter product helium (3He) concentrations during a previous sampling event in 1998 [9].

L123-125: How did you estimate porosity?

Porosity was assigned a standard value of 0.3 as field estimates were not available. This information is in the Limitations section, but it will be added earlier in the article in the Whiting Field Groundwater Model (WFGM) section.

L146-160: Add a figure showing horizontal and vertical model grid, boundary conditions, and discretization of the drain.

A new figure (fig. 4) is added to the revised manuscript showing the model and drains.

L164: Provide the data source for precipitation data and information about your precipitation data

The net recharge (precipitation – ET) was calibrated to produce the correct volumes in the model. This is clarified in a new sentence in the third paragraph of section 3.2. Whiting Field Groundwater Model (WFGM).

L165-167: Provide data source for GW table data or show you analysis steps

Reference [3] has been added to the table here as it has details on the groundwater measurement used in the development of the model.

L167-171: How did you estimate ET? Estimating both recharge and hydraulic conductivity simultaneously with inverse modeling without data to constrain either of them can lead to an ill-posed model.

In the reference document for the groundwater model [3] it discusses the use of a USGS streamflow gage to calibrate net recharge to an average recharge rate of 61.5 inches per year (in/yr), thus constraining the simulation with flow (and the measured groundwater heads). With an average precipitation of 69.5 in/yr in reference [2], this seems reasonable. As this information is in the cited reference document for the groundwater model, it is not repeated in this article but is mentioned in the third paragraph of section 3.2. Whiting Field Groundwater Model (WFGM).

L173-197: Optional: You provided a source for your lithologic and hydraulic conductivity analysis. It might be useful to show additional details and show an example for this analysis to make the paper self-contained (e.g., Elshall et al., 2013; Elshall & Tsai, 2014).

Although this article is focused on the particle-tracking application and not the groundwater model development in reference [3], a brief discussion has been added here about model parameter development methods to contrast with the method used here. These references are included in the revised manuscript.

L202-204: Before showing your simulated head, show first or provide a source for your calibration results (e.g., residuals vs, observed, simulated vs. observed, sum of squared residuals, etc.), estimate hydraulic conductivity field, estimated porosity, and the other model parameters that you calibrated.

The document Landmeyer et al, 2021 [3] describes the development of the groundwater flow model including calibration and the model fit. This article starts with this calibrated model and discusses the application of particle tracking. The following is added as the first sentence in section 3.2. Whiting Field Groundwater Model (WFGM): “Following is a general description of the WFGM; further details on model development, calibration, and results can be found in [3].”

L214 Figure 5: You might want to use different colors for four contours. For example, I cannot easily see the GW divide that you mentioned early with just one color?

We would like to point out that this figure is primarily for referencing the water table produced  by the model; details such as the groundwater divide are mentioned but not necessary for the particle-tracking results. The reader is directed to reference [3] as in the previous comment.

L235: 237: What is number of particles and the initial particle locations at each well (e.g., Elshall et al., 2020). Please state if results are sensitive to the initial practical locations and number.

Thank you for this comment. A test indicated low sensitivity to position of each particle, due to being a steady-state simulation, so a single particle at the depth of the screened interval were used in the test. This was chosen as the simplest scheme, and expanding the number of particles could still be an option. This is discussed in Simulation Limitations part 6, but a description is added here just before the Results section “This steady-state simulation has less spatial variability than a transient simulation, so a particle is backtracked from each well location at the depth of the screened interval and considered representative of that well.”

L261: “The time for particles to travel from the land surface” This is unclear. Do you mean from your water table? Are you also simulating the unsaturated zone?

Unfortunately, we misstated this in the original paper and this issue has now been corrected; it is the time for particles to travel from the top of the water table. The particle-tracking algorithm considers the water table as the upper boundary. The model does simulate the unsaturated zone, but the particle tracking does not cover this zone.

L271: What is the start and end dates of your simulations or this is a steady-state simulation? You assumed a constant pumping rate throughout the simulation. Please confirm.

The simulation is steady-state and the pumping rates are average values. Although we described this previously in the body text, it will also be included here before table 2.

L281-284L How would you explain the similarity between particle travel time with and without pumping?

The groundwater flow velocities are higher when pumping, which makes travel times shorter, but also groundwater is drawn from farther away, which lengthens travel times. So the travel times tend not to be dramatically different between pumping versus non pumping. Most travel times are longer with pumping due to the particle traveling farther. This is mentioned in the paragraph after table 2.

Elshall, A. S., Ye, M., & Finkel, M. (2020). Evaluating two multi-model simulation–optimization approaches for managing groundwater contaminant plumes. Journal of Hydrology, 590, 125427. https://doi.org/10.1016/j.jhydrol.2020.125427

Elshall, A.S., & Tsai, F. T.-C. (2014). Constructive epistemic modeling of groundwater flow with geological structure and boundary condition uncertainty under the Bayesian paradigm. Journal of Hydrology, 517. https://doi.org/10.1016/j.jhydrol.2014.05.027

Elshall, A.S., Tsai, F. T.-C., & Hanor, J. S. (2013). Indicator geostatistics for reconstructing Baton Rouge aquifer-fault hydrostratigraphy, Louisiana, USA. Hydrogeology Journal, 21(8). https://doi.org/10.1007/s10040-013-1037-5

Submission Date

31 October 2023

Date of this review

20 Nov 2023 16:41:50

© 1996-2023 MDPI (Basel, Switzerland) unless otherwise stated

Round 2

Reviewer 1 Report

Comments and Suggestions for Authors

Thank you for answering my comments.

Author Response

Thank you for your extensive review!

Reviewer 3 Report

Comments and Suggestions for Authors

Dear Authors,

I appreciate the effort you put into correcting this article. I have no further comments, all ambiguities have been corrected/supplemented in the new version of the manuscript or discussed as part of the Authors' responses. I am not entirely convinced of the concept of figures adopted by the Authors, but perhaps this is just a matter of a separate aesthetic approach to presenting information at figures - nulla ratio gustum. I hope that my comments have helped to improve this interesting work and that it will gain recognition in the eyes of the hydrological community. Good luck in your further scientific research.

Author Response

There certainly are multiple ways of presenting data in figures and most of our figures involve overlaying data on a simple base map. I'm sure that the alternatives could be considered, but we are going with the current format.